# Evidence for a universal association of auditory roughness with musical stability

**Andrew J. Milne**[1☉]*, **Eline A. Smit**[1,2☉], **Hannah S. Sarvasy**[1], **Roger T. Dean**[1]

**1** MARCS Institute for Brain, Behaviour and Development, Western Sydney University, Sydney, NSW, Australia, **2** Department of Linguistics, University of Konstanz, Konstanz, Germany

☉ These authors contributed equally to this work.
* a.milne@westernsydney.edu.au

**Data Availability Statement:** The data, materials, and models for all experiments are available at https://osf.io/c3e9y/ and the experiment was preregistered (https://osf.io/qk4f9).

## Abstract

We provide evidence that the *roughness* of chords—a psychoacoustic property resulting from unresolved frequency components—is associated with perceived musical stability (operationalized as finishedness) in participants with differing levels and types of exposure to Western or Western-like music. Three groups of participants were tested in a remote cloud forest region of Papua New Guinea (PNG), and two groups in Sydney, Australia (musicians and non-musicians). Unlike prominent prior studies of consonance/dissonance across cultures, we framed the concept of consonance as stability rather than as pleasantness. We find a negative relationship between roughness and musical stability in every group including the PNG community with minimal experience of musical harmony. The effect of roughness is stronger for the Sydney participants, particularly musicians. We find an effect of *harmonicity*—a psychoacoustic property resulting from chords having a spectral structure resembling a single pitched tone (such as produced by human vowel sounds)—only in the Sydney musician group, which indicates this feature's effect is mediated via a culture-dependent mechanism. In sum, these results underline the importance of both universal and cultural mechanisms in music cognition, and they suggest powerful implications for understanding the origin of pitch structures in Western tonal music as well as on possibilities for new musical forms that align with humans' perceptual and cognitive biases. They also highlight the importance of how consonance/dissonance is operationalized and explained to participants—particularly those with minimal prior exposure to musical harmony.

## Introduction

Consonance and dissonance (henceforth C/D) are amongst the most important principles governing the structure of Western tonal music: consonant chords such as major and minor triads are more common than dissonant chords, particularly at perceptually privileged locations (e.g., strong beats and phrase endings). For example, in such music, it would be uncommon for a musical phrase, let alone a whole piece, to end on anything other than a major or minor chord. Different types of harmonic organization also play important roles in different musical traditions; for instance, the prominence of harmonic major seconds in several two-part vocal traditions worldwide [1, 2], and the shimmering harmonies of inharmonic tones in

**Funding:** AJM is the recipient of an Australian Research Council (https://www.arc.gov.au) Discovery Early Career Researcher Award (project number DE170100353) funded by the Australian Government. EAS is the recipient of funds from a Western Sydney University (https://www.westernsydney.edu.au) Postgraduate Scholarship from the MARCS Institute for Brain, Behaviour and Development. HSS is the recipient of an Australian Research Council (https://www.arc.gov.au) Discovery Early Career Research Award (project number DE180101609), and an award from the Australian Research Council (https://www.arc.gov.au) Centre of Excellence for the Dynamics of Language, Australian National University (CE140100041). The funders had no role in study design, data collection and analysis, decision to publish, or preparation of the manuscript.

**Competing interests:** The authors have declared that no competing interests exist.

Indonesian gamelan, which are tuned to favour a beat frequency of about 8 Hz [3]. Furthermore, the tonal hierarchies found in several non-Western as well as Western traditions [4–8], and expert listeners' reliable detection of the most stable pitch in songs from the NHS Discography (86 audio recordings from an approximately representative sample of human societies) [9] also support the world-wide importance of such hierarchies.

Consonance and dissonance also influence Western listeners' affective responses to music: consonant chords are often considered pleasant or stable sounding, while dissonant chords are comparatively unpleasant or unstable; hence, behavioral studies examining C/D often ask participants to rate the pleasantness of musical stimuli (e.g., [10–12]) or their stability (e.g., [13]) (another historical characterization of C/D is 'fusion'; [14, 15]). The precise framing of the concept of consonance/dissonance—how it is explained to participants—is of crucial importance. Not only does this framing need to be understandable by all participants tested, it is also plausible that certain characterizations of C/D may reflect more fundamental and potentially universal aspects of its perception, while others may reflect more layered and cultural aspects.

Here, we will focus on the stability aspect of C/D (experimentally framed as how 'finished' a chord sounds), rather than pleasantness, for two reasons. (Note that other terms are closely associated with 'stability': 'tension', which implies an imminent 'release' or 'relaxation', hence the chord is unstable; 'closure' or 'finishedness', the focus of the current study, which both imply stability because no imminent change is expected.) First, Western pedagogical music theory's primary focus on C/D is how it relates to the music's metrical and structural organization (consonance on strong beats, dissonant chords 'passing' between them on the weak beats), not to their affective valence. Second, for Western listeners, chords that are generally considered highly 'dissonant' and unstable (e.g., semitone clusters) can be perceived as pleasant. Indeed, some non-Western musical cultures seek to maximize dissonance, particularly in two-part vocal textures [1, 2, 16]—perhaps because they æsthetically value the sense of motion implied by unstable chords. More generally, different listeners may agree on a description of a feature but differ on their æsthetic evaluation of it. Hence more descriptive terms such 'stability' or 'finishedness' are arguably more fundamental and widely applicable characterizations of C/D than are more valenced (positive/negative) terms such as 'pleasantness' or 'liking'. Indeed, this theoretical argument has been empirically supported: responses to the descriptive term 'tension' are less influenced by music-cultural exposure than are responses to the valenced terms 'pleasantness' and 'preference' [17].

Suggested mechanisms to account for common associations between chords and C/D (variously understood as stability, pleasantness, etc.), have arisen from quite different theoretical perspectives, ancient and modern: mystical/numerological [18, 19], psychoacoustical [14, 20–24], neurobiological [25, 26], or via iconic signification whereby the auditory stimulus resembles, perhaps cross-modally [27], the affect it signifies [28–31]; see also [32] for a recent review of proposed mechanisms for C/D. All the above putative mechanisms are *non-cultural* and hence *universal* because they apply similarly to any human, regardless of their cultural background, and result in non-arbitrary associations between audio features and affect.

An alternative explanation is that these differing consonances and dissonances are driven by cultural exposure—via *familiarity* (mere exposure; [33]) and *associative learning*. For example, a listener may feel a major chord is more pleasant simply because they have heard that chord more often in their culture's music [34]; they may feel a major chord is more stable because, through their cultural exposure, it is more consistently heard in stable musical positions, such as the ends of phrases or sections. Both these psychological mechanisms require knowledge of a (possibly arbitrary) musical structure that privileges certain chords. Hence these are cultural, not universal, explanations for consonance and dissonance.

To be more formal about our definitions of 'cultural', 'non-cultural', and 'universal' effects, let us label a participant's affective response as $Y$ (e.g., pleasantness or stability), a psychoacoustic feature as $X$ (e.g., roughness or harmonicity), the participant's culture as $C$, and a cultural mechanism as $M$ (e.g., the participant's long-term familiarity with that value of $X$ or their learned associations with that level of $X$, given their culture $C$). For a given stimulus, the value of the psychoacoustic feature $X$ does not differ between cultures (it may differ slightly between individuals with normal hearing but not in aggregate in a given culture) whereas, by definition, a feature's culture-dependent distribution $M$ may differ (e.g., in Western music, harmonic timbres are more common than inharmonic; in gamelan music, inharmonic timbres are more common). Causally, $M$ mediates the effect of $X$; as shown in Fig 1. The direct path from $X$ to $Y$ would be a *non-cultural* effect of $X$, the mediated path from $M$ to $Y$ would be a *cultural* effect of $X$. For any psychoacoustic feature $X$, both types of effect—non-cultural and cultural—can run in parallel (and may have different strengths); putative mechanisms were exemplified in the previous two paragraphs. The non-cultural effect would be universal (this follows from its definition), but a cultural effect could also be universal if the associated feature's distribution does not change between cultures; for example, human speech is a cultural universal and has high harmonicity, which may encourage a universal preference for chords with high harmonicity [30]. However, there would be no ethical way of experimentally changing long-term

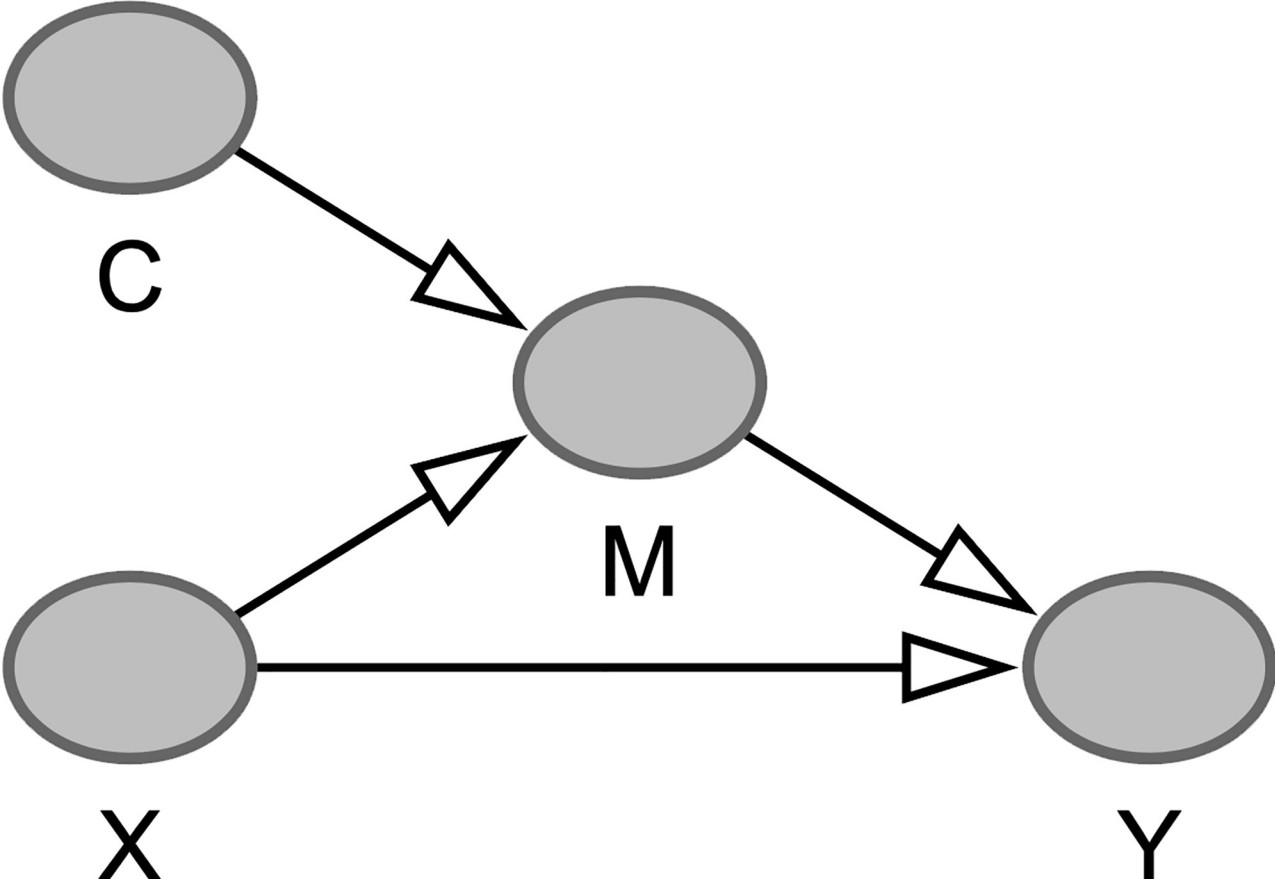

**Fig 1. A directed acyclic graph showing our causal model of how psychoacoustic ($X$, e.g., roughness) and cultural variables (culture $C$ and mechanism $M$, e.g., familiarity) influence affective responses ($Y$, e.g., stability).** The path from $X$ to $Y$ is a 'non-cultural' effect, hence 'universal'; the path from $M$ to $Y$ is a 'cultural' effect. However, if there is no variation in $M$ between cultures, a cultural effect can also be 'universal'.

exposure to speech, or another cultural universal, so as to uncover its precise effect on music-induced affect. We prefer the universal–cultural framing, also used in [35–39], over the biology/innate–culture framing in [29, 30, 40–43] because the latter may not be amenable to experimental testing.

Although it is theoretically possible that the perceived C/D of chords arises from effects that are 1) solely universal, 2) solely cultural, or 3) a combination, the first is improbable given ample experimental evidence in favour of cultural influences playing an important role [35, 36, 39, 42, 44]. The second is more plausible: a recent investigation with the Tsimanè—a remote Bolivian community with minimal exposure to Western music—found they did not rate intervals and chords deemed consonant in Western music theory as more pleasant [42]. This suggests perceptions of C/D may be a purely cultural effect. However, in that study, ratings were requested for the 'pleasantness' of different dyads and triads, not their 'stability' (or some other related term such as 'closure', 'tension', or 'finishedness'). (Bowling et al. [43] also raise several other possible methodological issues.)

Interestingly, recent experiments have shown that several psychoacoustic features (roughness, harmonicity, spectral entropy, and mean pitch, all defined subsequently), have—for Western listeners—effects on pleasantness ratings that generalize to unfamiliar microtonal chords (these are chords comprising intervals not used in Western music) [10, 45, 46]. *Roughness* refers to the rapid *beating* (undulations of loudness) that occurs when an audio signal has frequency components that are close enough that they cannot be separately resolved by the auditory system [21, 22]. *Harmonicity* quantifies the similarity, given inaccuracies of pitch perception, of the signal's spectrum to that of a single harmonic complex tone (a common example being a human vowel sound) [11, 24, 30, 47]. *Spectral entropy* quantifies the complexity, given inaccuracies of pitch perception, of the signal's spectrum [10, 48]. *Mean pitch* is simply the mean (in a log-frequency measure like semitones) of the notated pitches in the chord (i.e., the pitches as written, rather than directly extracted from the audio signal). The just-mentioned ability of these features to predict responses to unfamiliar stimuli is a necessary but not sufficient condition for universality, but still an important test to pass. Therefore, we tested the influence of these psychoacoustic features on our culturally diverse participants, with particular interest in their effect in a remote community in Papua New Guinea with minimal exposure to any kind of musical harmony (chords).

In the previously mentioned Bolivian study, participants with minimal exposure to Western music were apparently indifferent to the C/D of chords. However, the roughness and harmonicity of these acoustic signals were not directly measured or modelled; indeed, the roughness of an acoustic signal is highly dependent on subtle aspects—such as the loudness of every partial—which are not necessarily reflected by conventional musical notation [42]. Crucially, here we estimate the values of roughness, harmonicity, and spectral entropy directly from the acoustic signals heard by participants (i.e., by computational processing of the audio files used, not from their symbolic notational representation; detailed subsequently). Another important difference is that, for the reasons detailed earlier, we asked our participants to compare how 'finished' chords sound (a term related to 'stability') not their 'pleasantness'.

## Experiments in Papua New Guinea and Australia

We conducted the experiment in several remote villages in the Uruwa River Valley, Saruwaged Mountains, Morobe Province, Papua New Guinea (PNG). As subsequently detailed, these participants fall into three groups with differing levels and types of exposure to Western or Western-like music (we use these terms to refer any music that makes substantial use of a set of structural features commonly found in European music from at least the 17th century,

including frequent use of major and minor chords and diatonic scales; a full definition is provided in S1 Appendix). The same experiment was repeated in Sydney, Australia, with two cohorts: musicians and non-musicians. We operationalized C/D by asking participants to choose which of two chord progressions was the most 'finished'. As detailed below, this choice of terminology ('finished' rather than 'stable') was driven by the availability of appropriate terms in the language spoken in this area.

Three types of music are present in different, at times overlapping, parts of the Uruwa River Valley: traditional indigenous song; Western-influenced *stringben* (which means 'string band' in Tok Pisin, a PNG lingua franca); and Western hymns. Traditional indigenous songs are performed for specific occasions or gatherings or, in non-performance contexts, for personal pleasure. They are commonly accompanied with a *uwing* drum, which is a wooden hourglass-shaped drum with a single animal-skin head [49], or handmade flutes. We report a detailed analysis of six local recordings of traditional performances in [44]. In general, melodies are monophonic (sung solo or in unison). Melodic interval sizes are rarely larger than a perfect fifth, and generally do not conform with Western intervals. As confirmed by local knowledge, there are no obvious differences between music traditions in different villages of this region. There is no evidence that harmonic dyads or chords (beyond unisons and octaves) form a part of the Uruwa people's traditional music.

Missionary activity led villages to adopt either the Lutheran or the Seventh-day Adventist (SDA) church. However, within and between villages, the extent of involvement with either differs. Lutheran church service hymns are described by local people and literature on the Lutheran movement in urban areas of Morobe Province to be a mixture of traditional and *stringben* sung melodies, often accompanied by guitar or ukulele in the *stringben* style. *Stringben* songs are mostly in a major key with generally major chords [50, 51]. An analysis of local Lutheran hymns presented in [44] confirms this. SDA followers generally hear and perform hymns from the *SDA Hymnal* [52] at services twice a week. The hymns are mostly sung in English and, as analysed in [44], the majority of the songs are in a major key and use predominantly major chords. Lutherans in Kotet and Yawan villages lack their own Lutheran church (Kotet's was demolished in 2011) and are too far from the closest Lutheran alternative to make regular church attendance practicable. Both these villages have ready access to an SDA church, so SDA residents can attend church regularly. Historical information on the area is in S1 Appendix.

The above implies three groups of Uruwa participants with different levels or types of exposure to Western or Western-like music. The *minimal exposure group* consists of non-church-goers who would have had very limited and sporadic exposure to Western-like music for at least seven years prior to the experiment. The *Lutheran exposure group* consists of all Lutheran church-goers—they have regular exposure to major chords, but less exposure to minor. The *SDA exposure group* consists of all SDA church-goers who have regular exposure to major chords and, compared to the Lutheran group, a slightly wider palette of Western chords and less regular exposure to indigenous music than the other two groups. These groupings were based on three responses to the questionnaire—church attendance, village of residence, and church denomination. We use the same groupings in [44].

Both Sydney participant groups (musicians and non-musicians) have extensive exposure to Western music. Furthermore, the musicians, due to their greater exposure to and knowledge of Western music, should have better music perception skills compared to the non-musicians. Hence, we have five participant groups in our statistical analysis with differing levels and types of exposure to Western or Western-like music.

To examine C/D perception in these groups, participants were presented with many different pairs of chords (dyads and triads). In each trial, the participant heard one of these ordered

chord pairs followed by the same two chords but played in reverse order, and was asked which of the two pairs 'has finished' (*burettak* for the Uruwa participants—this Nungon word was chosen because it is the best available musically applicable approximation for 'stable'). The dependent variable is the probability of choosing the second chord pair as the 'finished' one. Based on the previously discussed theories and experimental results, we hypothesized this probability would be negatively related to the change in roughness (Δ*roughness*) and spectral entropy (Δ*spectral entropy*) from the penultimate chord to the final chord, and positively related to the analogous change in harmonicity (Δ*harmonicity*) and mean pitch (Δ*mean pitch*), with the strength of these effects changing by the community's level and type of exposure to Western or Western-like music and harmony (pre-registered report: https://osf.io/qk4f9). The presence or absence of each effect in the group with minimal exposure to Western music is of particular interest because it is indicative of the effect's universality.

## Methods

### Participants from the Uruwa River Valley

One-hundred seventy participants from the Uruwa River Valley, Saruwaged Mountains, Morobe Province, Papua New Guinea participated in the experiment, and 17,203 observations were collected ($n$ = 170, female = 85, mean age = 33.3 years, s.d. = 13.9 years). Interview data were missing from one of these participants, so that participant was removed from the analysis. Participants came from one of the following seven villages: Towet, Worin, Bembe, Mitmit, Mup, Kotet, or Yawan (see Fig 2). Participants were recruited, and the experiment conducted, in four locations over the following dates in 2019: Towet (27 Jun–7 Jul); Mup (14–25 Jul); Mitmit (30 Jul–4 Aug); Kotet (6–8 Aug and 1–3 Oct); Yawan (3–8 Oct). The PsychoPy script used for the experiment de-identified participants before saving the data. Their names, questionnaire responses, and PsychoPy codes were recorded by a research assistant in a separate spreadsheet, which was accessible to the authors. The field trip involved three other experiments (related to language), all running simultaneously. For this 'research fair', the entire village of Towet took two weeks off from regular duties to be available as participants, research assistants, cooks, and security for the researchers. Geographically, Towet and the other villages may be relatively close to each other, but the mountainous terrain means that walking between neighbouring villages is difficult (and possibly dangerous for non-locals) and can take between 30 minutes and a few hours for those familiar with the terrain. Due to this and to time constraints, the experiments in Mup, Mitmit, Kotet, and Yawan were conducted by local research assistants Namush Urung, Ben Waum and Nathalyne Ögate.

Test dates, location, and participant demographics are detailed in Table A1 in S1 Appendix. The sample size was chosen to be as large as possible depending on the availability of adults in the villages. Participants were rewarded with 50 PGK (about 20 AUD), which was the rate agreed in advance with community leaders as commensurate with the time (approximately 45 minutes) and effort involved (the same rate was agreed for every experiment in the 'research fair'). Written informed consent was obtained from all participants prior to the start of the experiment.

### Participants from Sydney

Nineteen musicians (female = 11, mean age = 27.3 years, s.d. = 8.7 years) and 60 non-musicians (female = 50, mean age = 21.8 years, s.d. = 6.4 years) were tested at Western Sydney University in Sydney, Australia. The experiment was conducted in 2019 from 6 Aug to 8 Oct; recruitment started one week before that and continued until completion. The online questionnaire and the PsychoPy script used for the experiment de-identified participants before

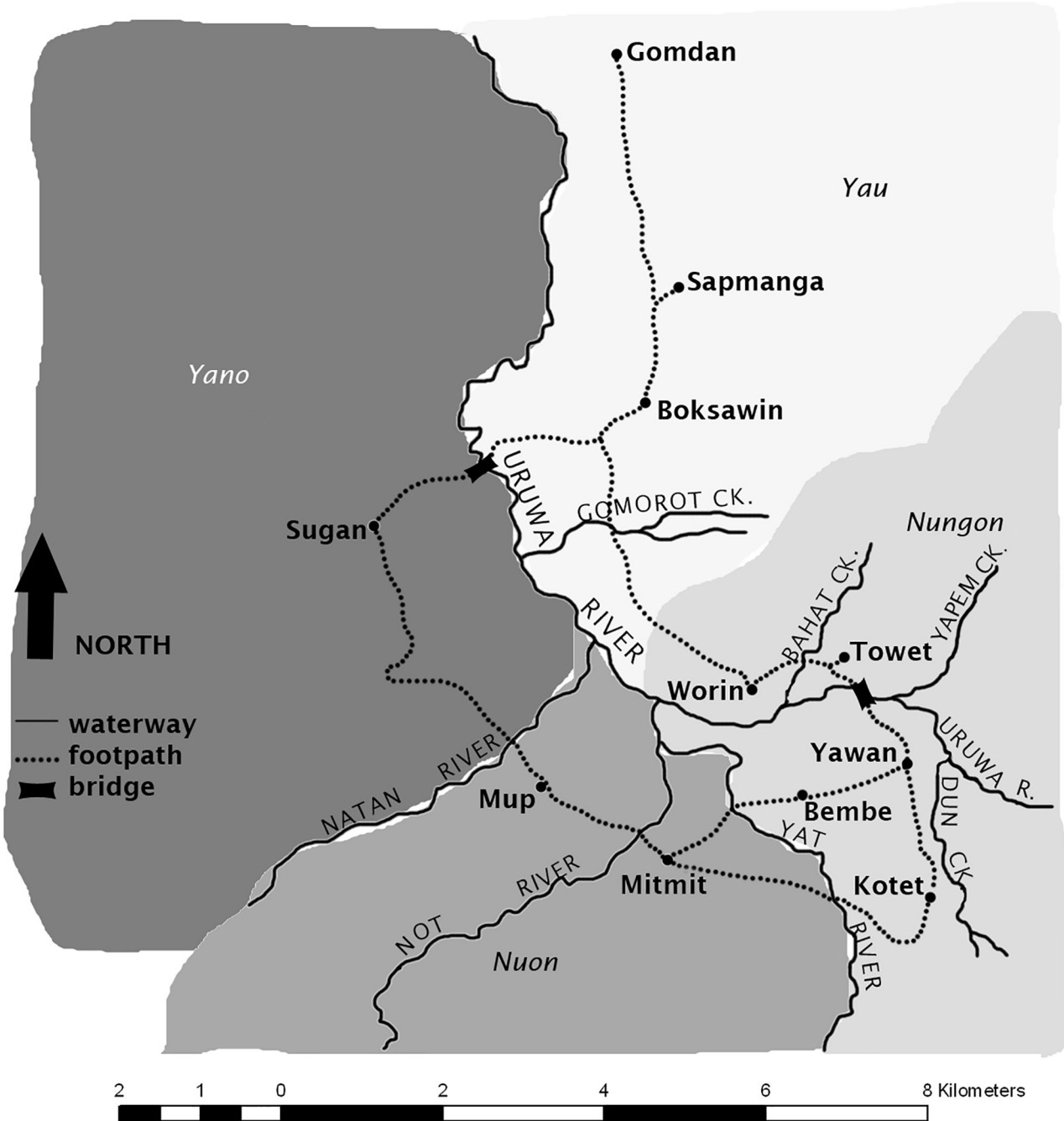

**Fig 2. Villages and languages (italics and shadings) in the area.** Figure from [44].

saving the data. Musicians were defined as those with at least five years of formal training in music, or equivalent musical experience, and they were recruited from the Sydney area by adverts and personal communication. They were rewarded with 30 AUD for their participation. In our studies, we typically use a smaller group of musicians compared to non-musicians, as their responses are generally more consistent [17] so even small effects are detectable in a cohort of this size. Non-musicians were undergraduate psychology students recruited through

the Western Sydney University's online research participation system (SONA), who received course credit for their participation. Of this latter cohort: 83% grew up in Australia, New Zealand, or a European country; 3% in the Middle East; 5% in South Asia; 3% in East Asia; 5% in Southeast Asia. The vast majority (98%) listen predominantly to Western music genres such as pop, classical, jazz, and R&B; only 10% listen to any non-Western music genres, and only 5% to genres with a possibility of distinctly non-Western harmony (Arabic, Chinese classical, Greek). All participants except one (the Arabic music listener) listen to at least one Western music genre. So, in terms of cultural heritage, this cohort has a reasonable amount of diversity but, in terms of musical exposure, they are relatively homogeneous and strongly focussed on music with conventional Western harmony. Written informed consent was obtained from all participants prior to the start of the experiment.

## Materials

Stimuli were generated in Max 7 (Cycling '74) and presented in PsychoPy v.3.0.2. Stimuli consisted of pairs of dyads, triads, cadences, and melodies and were created in a vocal and an instrumental timbre (only the dyads and triads are reported here). The vocal timbre was sampled from the voice sample library 'Voices of Rapture' (Soundiron) with the alto (A), tenor (T) and bass (B) singing 'Ah'. The instrumental timbre was sampled from 'Solo Strings Advanced' (Dan Deans) with the A as violin 2, T viola and B cello. Dyads use T and B, triads used A, T, and B. The MIDI was generated in Max 7, sent to the above sample sets played in Kontakt 5 (Native Instruments), individually panned left and right as typically found in a commercial recording, and saved as wav files. Stimuli were created in 12 different versions, counterbalancing between the two timbres per stimulus category and with random pitch transpositions, within one octave, for the dyads and triads, enabling us to test as many pairs of dyads and triads as possible. These 12 versions were varied between participants. Across participants and disregarding timbre and overall transposition, there were 130 different ordered pairs of triads and 82 different ordered pairs of dyads (each participant heard 60 of the former and 42 of the latter). Each participant's 144 stimuli were presented over five blocks. The stimuli can be downloaded from https://osf.io/c3e9y/.

The order of the blocks was as follows: dyads (30 trials), cadences (12 trials), dyads (30 trials), melodies (30 trials), triads (42 trials). The cadences and melodies blocks are not reported here because in those blocks we measured ratings of happiness, not finishedness; they were reported in a prior publication [44]. Block order was fixed, but stimulus presentation within blocks was randomized. Participants were asked to decide which stimulus induced happiness (one or two) for cadences and melodies, and which stimulus was 'finished' (one or two) for dyads and triads. The questions for this study were designed to accommodate the cultural and linguistic differences between Western listeners and PNG listeners.

In each trial in the dyad block, the participant would hear the Nungon word *ingguk* (which means 'one') followed by a sequence of two dyads (an ordered pair) followed by the word *yoi* ('two') followed by the same two dyads but played in reverse order. They were then asked to choose which of the two 'tunes' were 'finished' (*burettak*) (to which the answer was either *ingguk* or *yoi*). The word *burettak* was chosen as the best way to get at the concept of musical stability. The few other Nungon adjectives that might be said to relate to consonance reflect an æsthetic judgment, like *imbange* ('beautiful'/'good'/'nice') but, for the reasons detailed earlier, we preferred a word analogous to stability rather than to a valenced æsthetic affect. A forced (binary) choice was used because it is difficult for untrained listeners to quantify consonance in absolute rather than relative terms; it is generally easier to make an assessment relative to another stimulus. The design was also influenced by the use of the word *burettak* ('finished').

If a participant were to be presented with just two dyads, there would be a tendency to favour the second (*yoi*) as the more finished simply because it actually does finish the musical example. By instead presenting two ordered pairs of dyads (each pair preceded by *ingguk* or *yoi*), answers to the question will be less biased towards *yoi* (as can be seen in Fig 4).

All dyad *widths* (semitones between the two pitches) from 0 (unison) to 15 semitones (a minor tenth) were included and all ordered pairs of these were created such that their mean pitches differed by no more than 0.5 semitones and their widths differed by no more than 3 semitones. This results in a set of 130 ordered pairs (see S1 Appendix). As mentioned, in each trial, one of these ordered pairs of dyads would be followed by the same two dyads, but played in reverse order, and the participant would choose which of the two has 'finished'. From these 130, four subsets of 60 were selected, and each of these subsets was sounded with one of the two timbres, making eight distinct subsets. Every subset was randomly transposed, and each participant heard one of these subsets of 60 dyads.

The triad block had the same structure except triads (three-tone chords) were used instead of dyads. There were seven triad types (under the familiar $T_n$-type classification, which is invariant with respect to timbre, chord transposition, and individual pitch octave): major, minor, augmented, semitone cluster, a three-tone subset of a dominant seventh chord (root, third, seventh), a three-tone subset of a half-diminished seventh chord (root, fifth, seventh), and a seventh chord with no third (e.g., C–G–B♭). Most of these were presented in both closed and open positions (see S1 Appendix). Each participant heard one of two different sets of 42 different ordered pairs of these triad types (each set had two ordered pairs in common, so there was a total of 82 ordered pairs tested). Each ordered pair was available in five or six different transpositions, which were distributed into 12 between-participant groups. The pitches of the two triads in each ordered pair were chosen so as to minimize differences between the two triads' mean pitches.

## Procedure in the Uruwa River Valley

The field-research team (H.S.S., A.J.M., and E.A.S.) instructed the three local research assistants—Ben Waum, Nathalyne Ögate, and Namush Urung—how to operate the computers, start and run the PsychoPy experiment, and verbally explain the participant information sheet, consent form, and the experimental procedure to the participants. In Towet, N.U. and N.Ö. ran the experiments, independently, in the presence of one of the researchers (A.J.M. or E.A.S.) who observed quietly from a distance. In the absence of the research team, the experiments were conducted in the same way by B.W., N.U., and N.Ö. in Mitmit, Kotet, and Yawan, and by B.W. and N.U. in Mup. Further details are provided in Table A1 in S1 Appendix. The researchers and research assistants were unaware of the stimuli—the use of closed-back headphones ensured they were audible only for the participant, and stimulus ordering was randomized.

The local research assistants, and other community members who were involved in helping to organize the research trip, were paid for their time and work in PNG kina at rates agreed, in consultation with community leaders, to be fair and commensurate with the tasks performed (see [53, 54]). More details regarding the ethical considerations of the research are in S1 Appendix.

The experiment in Towet was conducted on the ground floor of a two-storey wooden building. The experiments conducted in the other villages were conducted in a quiet location. Participants were seated in front of one of the research assistants and listened to the stimuli through over-ear closed-back headphones (Audio-Technica M50x or Koss UR20). As outlined in the main text, every trial in the dyad or triad blocks followed the same procedure. First, participants heard the word *ingguk* ('one'), followed by stimulus 1, then the word *yoi* ('two'),

followed by stimulus 2. After stimulus presentation, participants were asked verbally, in Nungon: 'When you hear which tune, which one is finished?'. (Note that this sentence sounds odd in English, but it follows the correct grammatical structural of Nungon). Answers were recorded in PsychoPy by the local research assistants. Prior to the experiment, participants were presented with four practice trials with both questions to confirm understanding of the task. As detailed in the next subsection, the experiment was designed in collaboration with the community to ensure good task comprehension.

An interview in Nungon was conducted after the experiment to obtain more information about the participants' cultural and musical background (see S1 Appendix for the specific interview questions). Questions were asked orally in Nungon by local research assistants and responses were recorded by them or the researchers. The interviewers would interpret the responses as matching a particular choice of those available.

**Task comprehension.**   As with many cross-cultural adaptations of WEIRD (Western, educated, industrialized, rich, and democratic) research paradigms [55, 56], there are multiple components to comprehension of the task for our participants. We identified: (a) comprehension of the question being asked ('Is one or two finished?'); (b) familiarity with the notion that a melody could 'be finished'; and (c) understanding of the expected process a participant should go through to answer the question each time (evaluate the two musical fragments and choose the one that sounds finished to them).

(a). We have no doubt that participants understood the question at a simple linguistic level. H.S.S. is fluent in Nungon and wrote the authoritative 627-page reference grammar of the language [57]. The question was kept short and maximally simple to ensure that there was no ambiguity in it. H.S.S. discussed the wording of the question at length (in Nungon) with experiment fair organizers J.J., S.G., and L.Ö., and further with research assistants B.W., N.U., and N.Ö., to ensure that it was maximally felicitous. For 'finished', there were basically two alternative ways to express this: 'sung/said completely/fully', or 'finished'. We went with the second, which is clearer.

In sum, the question was succinct and well-formed in Nungon, structured as a typical short Nungon sentence. H.S.S. indicated that the wording left no room for ambiguity, and this was affirmed by the six consulting community members.

After such lengthy consultation with multiple community members, in Nungon, about the wording of the question, we did not expect participants to misunderstand it. This was borne out by the experimentation, during which we saw no indication that participants did not understand the question. They responded appropriately and promptly, 'one' or 'two'; we never observed anyone asking for the question to be repeated so that they could understand how to respond to it, or responding with something other than 'one' or 'two'.

(b). In H.S.S.'s experience, people in the Uruwa area do not habitually discuss music in detail. This could conceivably occur among practitioners of traditional music or stringben, but H.S.S. only recorded performances of these, not practice sessions. But the notion of 'finishing' speech is not strange; many people who recorded stories for H.S.S. over the years ended their stories by saying that the stories had finished, for instance.

We presented our plans for this research project in a Synergy seminar at the Summer Institute of Linguistics—PNG branch (in Ukuraumpa, Eastern highlands Province, PNG), and received helpful feedback from longtime fieldworkers in PNG, including Andy Grosh and Matt Taylor, on the best ways to frame the question. What could puzzle participants here would be not how a melodic fragment could be finished but how they would be able to decide that it had finished without knowing what song it comes from.

(c). Presumably, participants could understand the literal meaning of the question, and even be familiar with the notion being asked about (a musical piece 'finishing'), but still not understand how they were expected to respond to relatively rapid-fire questions and stimuli presentations in a psychology experiment.

It is widely known that there are cultural differences in the ways questions are used in discourse and in the expected ways of responding to them. Some Aboriginal communities in Australia, for instance, are known to have a cultural dispreference for asking and being asked direct questions, and to sometimes display so-called 'gratuitous concurrence' (replying 'yes' to positive questions and 'no' to negative questions, regardless of veracity; [58]) when interacting with non-Aboriginals. Our question did not use yes/no responses to positive/negative questions.

For the Uruwa community, participants lack familiarity with the notion of being asked the same question repeatedly about a series of musical fragments, with the accompanying expectation that they quickly evaluate each on its own merits. This lack of familiarity might not translate into difficulty; some participants might understand and enthusiastically take up the novel task. But the lack of familiarity could mean that other participants might respond at random, or use patterned responses, without understanding that they were not 'supposed to' do this in this type of experimentation.

## Procedure in Sydney

Participants in Sydney were tested in a soundproof room, seated behind a laptop and listened through headphones (Sennheiser HD 280 Pro, Beyerdynamic DT 770 Pro, or Sennheiser HD 650). As in PNG, the experimental task was explained by a research assistant, but participants indicated their answers themselves by pressing keys on the computer's keyboard. The same question was asked for Sydney participants as for the Uruwa participants, but translated to the English 'which one is finished?'. After the experiment, participants were asked to complete the Goldsmith's Musical Sophistication Index questionnaire, which is designed to elicit information from participants regarding their engagement with music, self-reported listening, singing skills, musical training, and their emotional engagement with music [59]. Questions related to their demographics, cultural heritage, and musical preferences were also asked.

Experiments were conducted in June/July 2019 (Towet) and July/August/October 2019 (other villages, Western Sydney University). The study was approved by the Towet community leaders, the Papua New Guinea National Research Institute, and the Western Sydney University Human Research Ethics Committee (H13179).

## Analytic strategies and modelling

The exposure group (minimal, Lutheran, SDA, Sydney non-musician, Sydney musician) encodes each participant's extent and type of exposure to Western music, and we are interested to see how this moderates the hypothesized effects of roughness, harmonicity, spectral entropy, and mean pitch, particularly for the minimal exposure group.

There are several different ways of mathematically representing the concepts of roughness and harmonicity [11] such that they can be extracted from audio. We used the Music Perception Toolbox because its calculations of these features have successfully predicted Western participants' ratings of the valence, pleasantness, and stability of both familiar Western chords and unfamiliar microtonal chords [10, 45, 60]. The Music Perception Toolbox's roughness calculations closely approximate those produced by the MIR Toolbox (see S1 Appendix) [61]; for harmonicity, it uses a straightforward mathematical method to quantify the similarity of the

spectrum to a harmonic complex tone [11, 47]. Spectral entropy is a recent measure [10, 48] available only within this toolbox. It is possible that different mathematical representations of these features might lead to different results; we did not test such alternatives in order to avoid inflating researcher degrees of freedom. The audio stimuli and data are open access so as to allow alternative analyses to be tested. The methods to calculate these features are detailed in S1 Appendix.

It is additionally worth noting that features such as these can also be (and indeed often are; e.g., [10–12]) calculated from *idealized* spectra such as can be obtained from assuming every notated pitch in a chord is a harmonic complex tone with an identical spectrum (relative to the fundamental). (To give an idea of the extent to which idealized and audio-based calculations may differ, if every chord used in the experiment is idealized by sets of tones with 36 harmonics with amplitude $1/h$, where $h$ is the harmonic number, the resulting correlations with the audio-based measures of roughness, harmonicity, and spectral entropy are, respectively, 0.85, 0.49, and 0.67.) This may be a useful approximation for Western listeners, who are familiar with numerous soundings of each chord type and so implicitly learn its average roughness, harmonicity, or spectral entropy; conversely, for participants without such exposure, the use of idealized spectra is inappropriate because, without previous exposure, participants would necessarily respond only to the actual sounds presented.

Before analysis, data were removed blockwise when a participant always answered 'one' or always answered 'two', or answered 'one' and 'two' alternately; such response patterns indicate that task instructions were not followed or that participants applied a specific strategy unrelated to their perception of finishedness and hence were removed, in line with the pre-registered report. A total of 27% of blocks were removed for the Uruwa participants. For the Sydney groups, no participants were removed but one block of triad trials was removed. (As summarized in S1 Appendix, an exploratory model fitted to all the data, with no blocks removed, has very similar posterior means and every strongly evidenced hypothesis reported in the next section remains strongly evidenced.) The resulting numbers of participants in each exposure group are: Uruwa minimal, 22 (19 for dyads, 21 for triads); Uruwa Lutheran, 39 (39 for dyads, 36 for triads); Uruwa SDA, 83 (80 for dyads, 66 for triads); Sydney non-musician, 60 (60 for dyads, 59 for triads); Sydney musicians, 19 (19 for dyads, 19 for triads). A more detailed breakdown is provided in Table A2 in S1 Appendix.

We used Bayesian multilevel logistic regression models to estimate the effects of the previously defined Δ*roughness*, Δ*harmonicity*, Δ*spectral entropy*, and Δ*mean pitch*, and their moderation by *exposure group* (Uruwa 'minimal', Uruwa 'Lutheran', Uruwa 'SDA', Sydney 'non-musician', or Sydney 'musician') and *trial type* ('dyads' or 'triads'). The latter factor was sum-coded to facilitate obtaining main effects for the other predictors, but we had no prior hypotheses about its effects. All continuous predictors were standardized except for mean pitch difference, which was left in semitone units to aid interpretability. All population-level effects were given weakly informative priors (a Student's *t*-distribution with a mean of 0, 3 degrees of freedom, and a scale of 1). This reflects our prior belief that the standardized effect size of each predictor is unlikely to be very large, and further serves to regularize (towards zero) any effects that may be only weakly informed by the data thereby decreasing the chance of false positives. The multilevel specification was structured to ensure all effects were partially pooled across participants but only within each exposure group (so there was no borrowing of information between exposure groups). Multicollinearity between the psychoacoustic predictors was low, and we used PSIS-LOO (an approximation of leave-one-out cross-validation; [62]) to compare all subsets of the four predictors. The best such model contained Δ*roughness*, Δ*harmonicity*, and Δ*mean pitch*, but not Δ*spectral entropy*; this is the model we report subsequently. More details on the modelling, tests for multicollinearity, and cross-validation can be found in S1 Appendix.

We use *posterior probabilities* and *evidence ratios* ('Post.p' and 'Evid.Ratio' in Table 1) to assess evidence in favour of directional hypothesis tests. For example, given a hypothesis that roughness reduces finishedness, we use the posterior probability that the effect of roughness is less than zero to quantify our certainty that this hypothesis is correct. Let us imagine that posterior probability is 0.89; this would mean that (given the model) there is a probability of 0.89 that roughness has a negative effect, and a probability of $1 - 0.89 = 0.11$ it has a positive effect. This can also be quantified as an evidence ratio, which are the odds in favour of the hypothesis so, for this imaginary example, the evidence ratio that roughness has a negative effect would be $0.89/0.11 = 8.09$. For a directional hypothesis like this, an evidence ratio $> 19$ (a posterior probability $> 0.95$) is somewhat analogous to a $p$-value $< 0.05$ [63, 64], and we refer to such a ratio as 'strong evidence'; for a bidirectional exploratory hypothesis, this threshold is $> 39$ (a posterior probability of 0.975 for one direction and 0.025 for the opposite direction, hence the total probability is still 0.95).

We do not use Bayes factors to test evidence for and against a point-null hypothesis, instead preferring the *region of practical equivalence* (ROPE) approach recommended by [65], which is to consider a range of effect sizes that are so small, they can be considered practically equivalent to zero. An advantage of this approach is that it is substantially less sensitive to the prior specification than are Bayes factors [65]; it also allows domain expertise to be used to determine a plausible region of practical equivalence. For our predictors, we define the ROPE as being in the interval $[-0.036, 0.036]$ (the standard deviations of the roughness, harmonicity and mean pitch predictors are, respectively, 1, 1, and 0.52; see 'Discussion' for qualification of effect sizes). For readers unfamiliar with Bayesian inference, in S1 Appendix we also summarize results from a simplified frequentist version of the main explanatory model (the results are substantively identical).

**Table 1. Hypothesis tests and summaries of the main effects (across dyads and triads) of Δ*roughness*, Δ*harmonicity*, and Δ*mean pitch*, in the five groups of participants.** 'Mean', 'Q5%', and 'Q95%' are the mean and 90% equal-tailed credibility interval for the logit-scale effect of a standard deviation increase in roughness and harmonicity, and a one-semitone increase in mean pitch. 'Evid.ratio' is the odds the effect is in the direction specified by the hypothesis, while 'Post.p' is its associated posterior probability. 'ROPE' is the probability the effect is practically equivalent to zero [65], which we define as being in the interval $[-0.036, 0.036]$.

| Grp by hypothesis | Mean | Q5% | Q95% | Evid.Ratio | Post.p | ROPE |
|---|---|---|---|---|---|---|
| *Effect of Δroughness < 0* | | | | | | |
| Uruwa: Minimal | −0.13 | −0.23 | −0.02 | 39.08 | 0.98 | 0.10 |
| Uruwa: Lutheran | −0.09 | −0.16 | −0.02 | 44.35 | 0.98 | 0.16 |
| Uruwa: SDA | −0.10 | −0.15 | −0.04 | 587.24 | 1.00 | 0.06 |
| Sydney: Non-mus | −0.29 | −0.36 | −0.21 | >19999.00 | 1.00 | 0.00 |
| Sydney: Musician | −0.85 | −1.06 | −0.66 | >19999.00 | 1.00 | 0.00 |
| *Effect of Δharmonicity > 0* | | | | | | |
| Uruwa: Minimal | −0.04 | −0.15 | 0.07 | 0.35 | 0.26 | 0.43 |
| Uruwa: Lutheran | −0.08 | −0.15 | 0.00 | 0.05 | 0.05 | 0.23 |
| Uruwa: SDA | −0.01 | −0.07 | 0.06 | 0.77 | 0.44 | 0.73 |
| Sydney: Non-mus | 0.05 | −0.01 | 0.12 | 9.01 | 0.90 | 0.42 |
| Sydney: Musician | 0.15 | 0.01 | 0.30 | 22.89 | 0.96 | 0.09 |
| *Effect of Δmean pitch > 0* | | | | | | |
| Uruwa: Minimal | −0.10 | −0.30 | 0.10 | 0.23 | 0.19 | 0.20 |
| Uruwa: Lutheran | 0.11 | −0.02 | 0.24 | 11.35 | 0.92 | 0.17 |
| Uruwa: SDA | −0.03 | −0.13 | 0.06 | 0.41 | 0.29 | 0.49 |
| Sydney: Non-mus | −0.15 | −0.25 | −0.05 | 0.01 | 0.01 | 0.05 |
| Sydney: Musician | −0.03 | −0.33 | 0.27 | 0.77 | 0.44 | 0.21 |

Additionally, for descriptive and illustrative purposes, we ran a Bayesian multilevel Thurstone (paired comparisons) model to estimate the relative stabilities of the different chord types for each of the exposure groups. As with the main model, all population-level effects had weakly informative priors centred at zero, and the multilevel structure ensured information was partially pooled between participants but only within each exposure group.

## Results

Fig 3 illustrates the relative stabilities of the chords as estimated by the 'descriptive' chord-type model. The chords (dyads and triads) are classified by $T_n$-type [66], which is common in Western music theory: this classification ignores timbre, chord transposition, and individual pitch octave and is, therefore, equivalent to conventional chord descriptions such as 'major', 'minor', 'diminished', and so forth. For example, this means that the following chords are all classified as (0, 4, 7) or 'major': (C4, E4, G4), (D4, F♯5, A6), (F4, A3, C4). The resulting visualization should, therefore, be straightforward to interpret for a person familiar with a modicum of Western music theory. Different music-theoretical classifications are shown in Figs A2 and A3 in S1 Appendix.

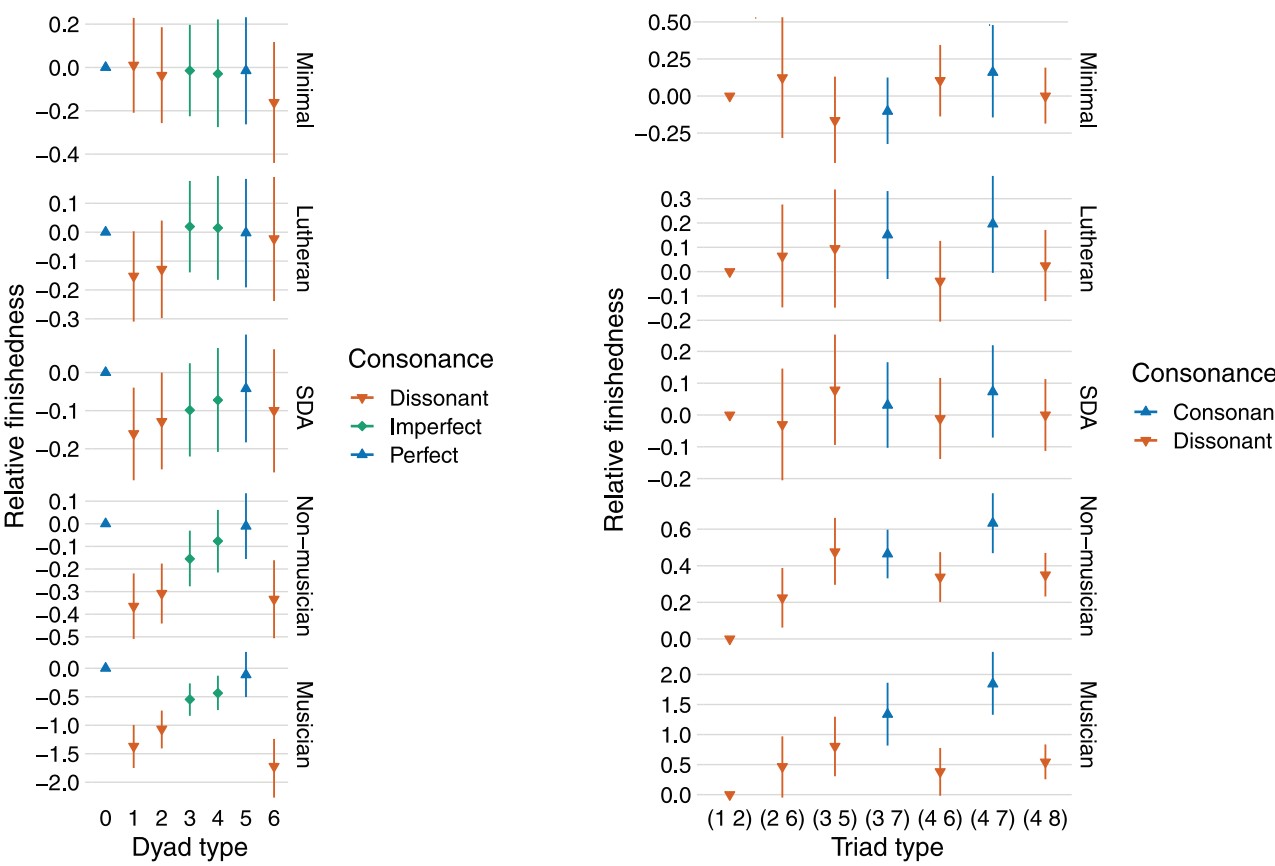

**Fig 3. The relative finishedness of every chord as estimated with a multilevel Bayesian Thurstone model [67] (this model is detailed in S1 Appendix).** The dyads and triads are classified by $T_n$-type, which disregards timbre, transposition, and all pitches' octaves. Each dyad's finishedness is relative to the unison's; each triad's finishedness is relative to the semitone cluster's. The units of the vertical axis are standard deviations on the normally distributed latent scale, which the Thurstone model assumes to underlie responses. The vertical scaling changes between the exposure groups to make any patterns easier to see (versions with identical scaling are available in Fig A3 in S1 Appendix). Conventional Western music-theoretical categorizations of consonance/dissonance are indicated by colour and shape. Each bar shows the 95% Bayesian credibility interval (equal-tailed) of the effect's posterior distribution; the central shape (triangle or diamond) shows its mean.

The pattern of responses by dyad-type is broadly consistent between the Sydney groups and the SDA and Lutheran groups, although very different in terms of size—the range of the Sydney musician group's responses is about five times greater than the non-musician group's, and about ten times greater than the Lutheran and SDA groups' responses. For the minimal exposure group, the only noticeable similarity to the other groups is the possibly lower perceived finishedness of the 6-semitone (tritone) dyad; it is notable that, for this exposure group, the semitone and whole-tone dyads are rated no less stable than the unison. By triad-type, the two Sydney groups' responses have a similar pattern but, as with dyads, the musician group's range is substantially (about three times) greater than the non-musician group's. Between all five groups, the triad-type patterns are less consistent, although the estimated means for (3, 5), (3, 7), and (4, 7) are the highest in every group except the minimal exposure group, and the estimated mean for the major triad (4, 7) is highest every group except SDA (where it is a close second).

However, the implicit assumptions made by conventional chord classifications may obscure underlying patterns that emerge when timbre, transposition, and octaves are no longer glossed over. The hypothesis driven model shows, for each exposure group, to what extent reported finishedness can be predicted by the previously detailed psychoacoustic features (roughness, harmonicity, spectral entropy), all of which are calculated directly from the audio signal and hence sensitive to timbre (more precisely, spectral content), chord transposition, and pitch octave. The remaining feature, mean pitch, is sensitive to transposition and inversion, but not timbre.

This model's results are summarized in Table 1 and visualized in Fig 4. For the main effects (across dyads and triads), we find very strong evidence that roughness plays a role—in the expected direction—in every cohort; notably, in the cohort with only limited and sporadic exposure to Western-like music in the preceding seven years. Conversely, harmonicity is only strongly evidenced—in the expected direction—for Sydney musicians. There is weak evidence for a positive effect in Sydney non-musicians. There is no evidence for a positive effect of harmonicity in any of the PNG groups; indeed, there is some evidence for a negative effect in the Lutheran group. Mean pitch has a strongly evidenced effect only in the Sydney non-musician group and in the opposite direction to that hypothesized. There is no strong evidence for any effect being practically equivalent to zero in any group.

The effects of roughness, harmonicity, and mean pitch are not consistently influenced by the use of dyads versus triads; indeed, for this bidirectional hypothesis, there is strong evidence for a difference only for roughness and harmonicity in the SDA group (for roughness, the estimated effect is −0.18 for dyads and −0.02 for triads; for harmonicity, −0.08 for dyads, 0.07 for triads). A plot showing the dyad and triad results separately is in Fig A4 in S1 Appendix.

It is useful to compare these results with those obtained from different blocks of the same experiment, which were reported in [44]. In each trial of those blocks, participants heard a pair of *cadences* (chord progressions) or *melodies*, and chose which was the *happy* one. Participants' responses were modeled by major versus minor and mean pitch (for the cadences) and mean pitch (for the melodies): the Lutheran and SDA groups showed attenuated Western-like responses to these, but there was no evidence for any effects in the minimal exposure group. This substantiates the presence of Western music-enculturation in the Lutheran and SDA groups but, as expected, not in the minimal exposure group. As reported in this article, the findings for chord finishedness are quite different because there is strong evidence for an effect of roughness in all groups, including the minimal exposure group which, due to its absence of recent exposure to any form of musical harmony and demonstrated lack of enculturation to Western music, is indicative of a universal effect of psychoacoustical roughness on perceived musical stability (operationalized as finishedness).

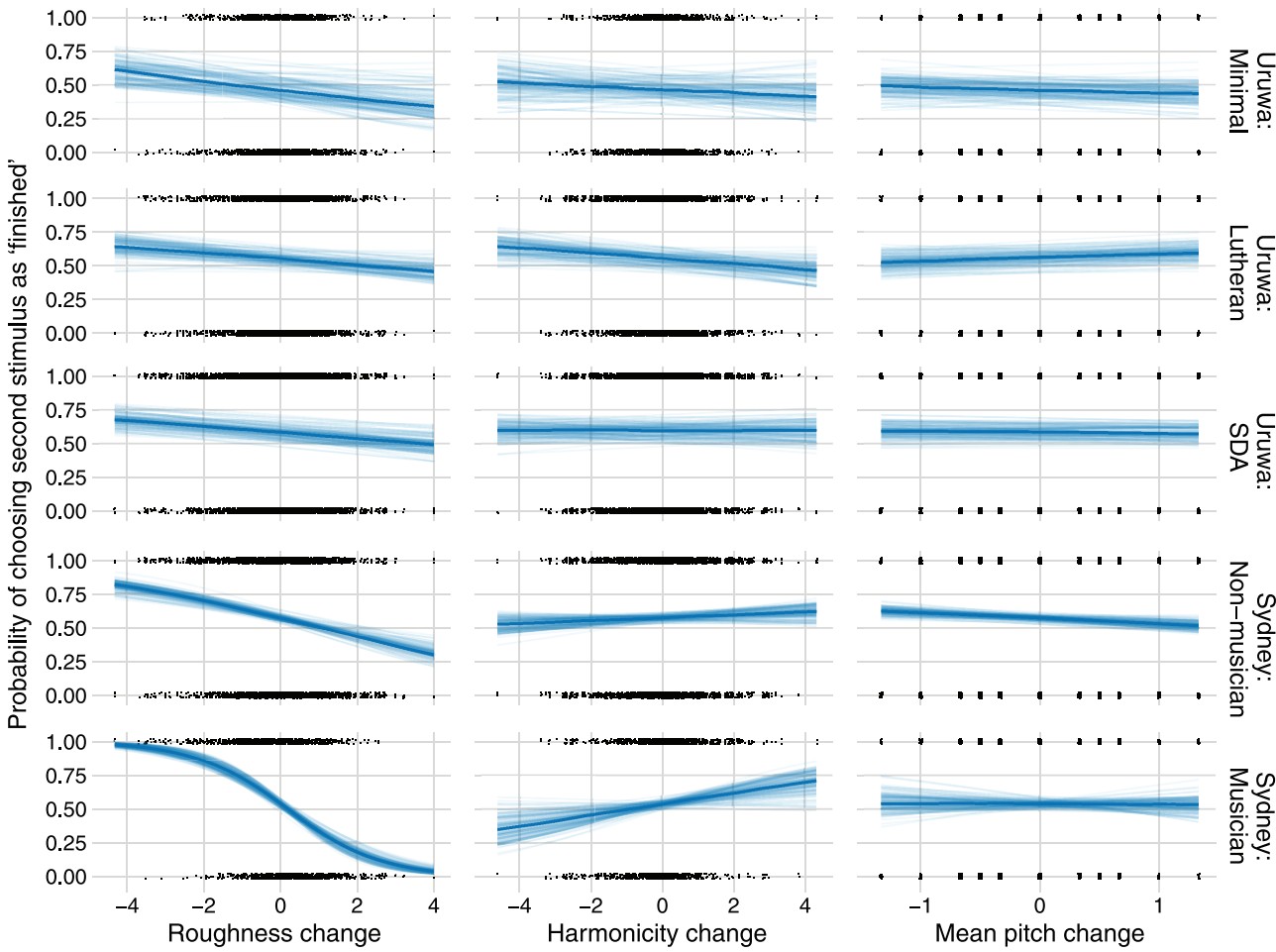

**Fig 4. Predicted effects for the five exposure groups.** Each trial had two chord pairs: the first pair was preceded by *ingguk* ('one'); the second pair by *yoi* ('two'). The second chord pair was the same as the first except in reverse order (hence the first and fourth chords are the same; the second and third chords are the same). After hearing these, the participant chose which of the two pairs sounded 'finished'. The plots show the probability of choosing the second chord pair as the 'finished' one as predicted by the Δ*roughness*, Δ*harmonicity*, and Δ*mean pitch* from the third chord to the fourth chord (all other predictors at their mean). Note that, for each predictor, because its change from the third to fourth chord is the negative of its change from the first to the second chord, it is sufficient (and possible) only to use one of these changes as a predictor. The units of Δ*roughness* and Δ*harmonicity* are their standard deviations, the units of Δ*mean pitch* pitch are semitones. For each plot, the posterior mean is shown with the thick line, while its uncertainty is visualized with 100 samples from the posterior distribution. Each black dot at the bottom or top is an instance of a trial where a participant answers 'one' or 'two', respectively.

## Discussion

Every group, including the PNG participants with minimal exposure to Western musical harmony, showed a negative effect of roughness on perceived stability (operationalized as finishedness). This provides evidence of a non-arbitrary and cross-cultural association between roughness and dissonance. Such an association may arise because roughness is the result of acoustic elements (frequencies) that cannot be perceptually resolved, creating a sensory confusion that urges resolution. Alternatively, it may be because the fast undulations of loudness (the defining feature of roughness) iconically signify—through cross-modal similarity [29]—physical motion that is fast and erratic, hence unstable and unfinished.

At first sight, the roughness effect sizes observed in every PNG community seem small (about 0.1 standard deviations). However, it is important to note that a composer or performer

has many ways to choose sequences of chords differing by a good number of standard deviations of roughness. As shown in Fig 4, there are many stimuli differing by approximately five standard deviations and some by about eight. Hence these 'small' effects can be quite meaningful: if the roughness of two chords differ by five standard deviations, an effect of 0.09 (as in the Lutheran group) equates to a probability of 0.61 that—in comparison to the rougher chord—the less rough chord will be heard as finished (because $\text{logit}^{-1}(5 \times 0.09) = 0.61$); a difference of eight standard deviations with an effect of 0.13 (as in the minimal exposure group) equates to a probability of 0.74 ($\text{logit}^{-1}(8 \times 0.13) = 0.74$). So large effects on stability perception are readily attainable.

Previous research has shown harmonicity to be a useful predictor of Western listeners' C/D ratings [11, 12], and one which generalizes to unfamiliar microtonal chords [10]. Here, crucially, we find that it does not generalize to participants who have less formal exposure to Western music than do the Sydney musician group. This aligns with prior research showing the effect of harmonicity (but not roughness) to increase with Western musical expertise [10, 24, 38]. This strongly suggests that although people are universally sensitive to harmonicity (e.g., the Tsimanè participants can discriminate inharmonic and harmonic tones; [42]), the association between harmonicity and stability is arbitrary and is learned. For example, a Western listener will implicitly learn that chords with high harmonicity (notably major chords) are typically used on strong metrical beats and at the ends of phrases. Through this culture-dependent learned association, harmonicity comes to symbolically signify stability (or, perhaps, pleasantness). However, unlike roughness [68] and stability, the psychological association between harmonicity and stability appears to be arbitrary: one can plausibly imagine a counterfactual universe with a musical culture that arbitrarily privileges inharmonic chords and that people in this universe would—through cultural learning—make the opposite psychological association: sounds with higher harmonicity are less stable (or less pleasant). This is less plausible for roughness, because its non-arbitrary underlying association with stability goes in the opposite direction and would need to be suppressed and supplanted by cultural mechanisms.

Our pre-registered directional hypothesis about mean pitch difference—that ascending sequences would sound more stable—was weakly motivated. It was based on Western listeners associating higher mean pitch with greater pleasantness or happiness [10] whereas here, we are testing stability. In the single group (Sydney non-musicians) where we observe strong evidence for an effect, it is actually in the opposite direction. Spectral entropy has also played a predictive role in ratings of pleasantness and happiness of microtonal chords [10]. Here, we find it does not have predictive utility beyond that provided by roughness and harmonicity (it is correlated with both): the model without spectral entropy performed slightly better under cross-validation.

Some limitations must be noted. First, as reported in [44] and previously encountered by [53], even though the experiment was designed in collaboration with members from the community and taking the local language and culture into account, participants in the Uruwa River Valley are not familiar with experimental settings. Thus, task comprehension may have varied slightly across the PNG and Sydney groups. That said, we designed the experiment in a manner that would be as meaningful as possible for participants both in Sydney and in PNG and this is reflected in the results. If participants did not understand the task at all, this would have led to random results; instead, we see strong evidence for an effect of roughness in every group. Second, the extent of participants' differing exposures to both local and non-local musical traditions and culture could be further and more thoroughly studied. Given that exposure early in childhood may shape perceptual experiences, a follow-up study may look more into how exposure to Western music throughout the lifespan may impact responses.

An obvious question that arises is why our results apparently differ so substantially from those of [42]. The most likely explanation is down to the key differences between our experimental designs and analyses. First, we asked about stability, not pleasantness. Secondly, we used a more contrasting set of chord types (e.g., including the semitone cluster). Thirdly, we measured roughness and harmonicity directly from the acoustic signals and used those to predict the responses, rather than conventional Western categorizations of the consonance or dissonance of symbolically characterized dyad and triad types (minor second, major third, major triad, augmented triad, etc.). The contrast between the findings of the chord-type analysis (Fig 3) and the psychoacoustic analysis (Fig 4) corroborate this point and, for participants with low exposure to Western music, highlight the importance of audio over symbolic analyses.

The stronger effects for the Sydney participants (particularly musicians)—for both roughness and harmonicity—clearly indicate the importance of cultural learning for, respectively, reinforcing and creating associations between psychoacoustic features and perceived musical stability. The presence of a distinct effect of roughness in even the most musically remote PNG community suggests a universal, or at least non-arbitrary, association between roughness and instability. Given this, it would be reasonable for roughness to be exploited by Western composers and musicians working with harmony (particularly during the medieval development of early polyphony, which introduced a number of previously unused, hence unfamiliar, 'imperfect' harmonic dyads). In so doing, many previous generations of Western music composers have successively reinforced roughness's influence on Western music listeners by making lower roughness chords more prevalent (hence, more familiar) at important structural locations such as phrase endings. These findings, therefore, provide a partial explanation for why Western music has privileged certain chords over others and for some of the affects—instability/tension versus stability/finishedness/release—induced by different chords.

The findings for roughness and harmonicity also have implications for novel musical forms: when using a microtonal tuning system with intervals that are not close to just intonation (such as 5-tone equal temperament), spectral re-tuning can be used to sacrifice harmonicity for lower roughness [47, 69, 70]. To the extent that the Western preference for harmonicity is a purely cultural—hence mutable—phenomenon, whereas roughness is not, this trade-off would seem to be a particularly useful musical strategy for the creation of novel music.

## Supporting information

**S1 Appendix. Appendix for 'Evidence for a universal association of auditory roughness with musical dissonance'.**
(PDF)

## Acknowledgments

We thank Nathalyn Ögate, Namush Urung, and Ben Waum for their assistance with data recording and interviewing the participants in Papua New Guinea; Farrah Sa'adullah for her assistance with data collection in Sydney and musicological analysis of the SDA hymns; Felix Dobrowohl for musicological analysis of the *stringben* recordings from Towet; Don Niles and Michael Webb for help with the ethnomusicology background; and the audience at the Summer Institute of Linguistics Synergy Lecture, especially Andy Grosh and Matthew Taylor, for providing feedback. Lyn Ögate, Stanly Girip, and James Jio were the highly efficient organizers of the Towet 'research fair' of which this experiment formed a part. Many thanks to everyone in the local communities involved with hosting us and making us feel welcome and safe.

## Author Contributions

**Conceptualization:** Andrew J. Milne, Eline A. Smit, Roger T. Dean.

**Data curation:** Andrew J. Milne, Eline A. Smit.

**Formal analysis:** Andrew J. Milne.

**Funding acquisition:** Andrew J. Milne, Hannah S. Sarvasy.

**Investigation:** Andrew J. Milne, Eline A. Smit, Hannah S. Sarvasy.

**Methodology:** Andrew J. Milne, Eline A. Smit, Hannah S. Sarvasy, Roger T. Dean.

**Project administration:** Andrew J. Milne, Hannah S. Sarvasy.

**Software:** Andrew J. Milne.

**Supervision:** Andrew J. Milne, Roger T. Dean.

**Validation:** Andrew J. Milne, Hannah S. Sarvasy.

**Visualization:** Andrew J. Milne.

**Writing – original draft:** Andrew J. Milne.

**Writing – review & editing:** Andrew J. Milne, Eline A. Smit, Hannah S. Sarvasy, Roger T. Dean.

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
