## [Decision Letter · Decision Letter 0]

26 Jun 2023

PONE-D-23-12398Evidence for a universal association of auditory roughness with musical dissonancePLOS ONE

Dear Dr. Milne,

Thank you for submitting your manuscript to PLOS ONE. After careful consideration, we feel that it has merit but does not fully meet PLOS ONE’s publication criteria as it currently stands. Therefore, we invite you to submit a revised version of the manuscript that addresses the points raised during the review process. As you can see, the reviewers both felt very positively about the manuscript, as did I. Most of the suggestions are to add clarity and consider some definitional distinctions that are made, comparing to other types of distinctions made in the literature, which will hopefully expand the audience for this paper.

We look forward to receiving your revised manuscript.

Kind regards,

Jessica Adrienne Grahn

Academic Editor

PLOS ONE

2. lease include a complete copy of PLOS’ questionnaire on inclusivity in global research in your revised manuscript. Our policy for research in this area aims to improve transparency in the reporting of research performed outside of researchers’ own country or community. The policy applies to researchers who have travelled to a different country to conduct research, research with Indigenous populations or their lands, and research on cultural artefacts. The questionnaire can also be requested at the journal’s discretion for any other submissions, even if these conditions are not met.  Please find more information on the policy and a link to download a blank copy of the questionnaire here: https://journals.plos.org/plosone/s/best-practices-in-research-reporting. Please upload a completed version of your questionnaire as Supporting Information when you resubmit your manuscript.

3. We note that Figure 2 in your submission contain copyrighted images. All PLOS content is published under the Creative Commons Attribution License (CC BY 4.0), which means that the manuscript, images, and Supporting Information files will be freely available online, and any third party is permitted to access, download, copy, distribute, and use these materials in any way, even commercially, with proper attribution. For more information, see our copyright guidelines: http://journals.plos.org/plosone/s/licenses-and-copyright.

b.If you are unable to obtain permission from the original copyright holder to publish these figures under the CC BY 4.0 license or if the copyright holder’s requirements are incompatible with the CC BY 4.0 license, please either i) remove the figure or ii) supply a replacement figure that complies with the CC BY 4.0 license. Please check copyright information on all replacement figures and update the figure caption with source information. If applicable, please specify in the figure caption text when a figure is similar but not identical to the original image and is therefore for illustrative purposes only.

Reviewers' comments:

Reviewer's Responses to Questions

**Comments to the Author**

1. Is the manuscript technically sound, and do the data support the conclusions?

Reviewer #1: Yes

Reviewer #2: Partly

2. Has the statistical analysis been performed appropriately and rigorously? 

Reviewer #1: Yes

Reviewer #2: Yes

3. Have the authors made all data underlying the findings in their manuscript fully available?

Reviewer #1: Yes

Reviewer #2: Yes

4. Is the manuscript presented in an intelligible fashion and written in standard English?

Reviewer #1: Yes

Reviewer #2: Yes

5. Review Comments to the Author

Reviewer #1: Dear authors, I read with very much interest your manuscript on how roughness is perceived across individuals with different exposure to Western sonorities. The research design is clear and well presented. The findings are relevant for the literature on the cross-cultural investigation of consonance and dissonance perception. There is only one point that I would like you to think about in the revision of the paper, which relates with the way in which you seemingly conceive the dichotomy between culture and nature. You will find more details in the following list of suggestions, together with a couple of recent important references that are missing.

1) The recent review on C/D by Di Stefano, Vuust, and Brattico (2022, Physics of Life Reviews) is relevant both as general reference for the introduction and, more importantly, for the discussion.

2) Is fusion also worth mentioning as a proxy for consonance in the introduction?

3) Please consider adding “harmonic” before “intervals” when introducing dyads on page 2 second paragraph. Again, regarding the putative mechanisms underlying C/D please refer to Di Stefano et al., 2022 (and probably expand the paragraph a bit).

4) “A listener may feel a major chord is more pleasant simply because they have heard that chord more often in their culture’s music”: please support this claim, as the link between exposure and pleasantness is not self-evident (e.g., Zajonc, R. B. (1968). Attitudinal effects of mere exposure. Journal of personality and social psychology, 9, 1.).

5) As observed in the general comment above, it is assumed that there is an obvious alternative between cultural vs universal origins of C/D. I know that this is often the way how the issue is presented in the debate. However, I don’t think these two terms have to be taken as opposites. Something cultural can be well universal, if we mean with universal “universally widespread across cultures and individuals”. Aren’t music and artistic practices themselves universal and cultural at the same time?

When reporting on the study by McDermott et al. (2016), you wrote that it seems to suggest a purely cultural mechanism for C/D. In similar occurrences, it is not entirely clear what “purely cultural” means. Moreover, what “for C/D” refers to in the previous sentence? Discrimination? Perception? Preference? Use? Please also mention the papers that raised issues regarding the nature paper (e.g., Bowling, D. L., Hoeschele, M., Gill, K. Z., & Fitch, W. T. (2017). The nature and nurture of musical consonance. Music Perception: An Interdisciplinary Journal, 35(1), 118-121.)

6) “In our studies, we typically use a smaller group of musicians compared to non-musicians, as their responses are generally very consistent and even small effects are detectable in a cohort of this size” (p. 5). I agree but probably you need to support the claim.

7) “Of those, only one participant (the Arabic music listener) did not indicate listening to any Western music genre.” Not sure I understand the meaning of this sentence.

8) P. 9: “A total of 27% of blocks were removed for the Uruwa participants.” It seems a significant portion, can you provide some comments on this?

9) Discussion, line 2: I would avoid the use of the (umbrella) term ‘natural’ here. How about ‘consistent’?

10) When discussing roughness, you might consider the recent review by Di Stefano, N., & Spence, C. (2022). Roughness perception: A multisensory/crossmodal perspective. Attention, Perception, & Psychophysics, 84(7), 2087-2114.

11) “the association between harmonicity and stability is not a natural one but is learned”, p. 13. Again, I would recommend rephrasing similar claims, as they seem conceptually problematic. What we learn can be natural (e.g., language, walking…). At the same time, universal traits are not necessarily natural, can be learned… In general, in the discussion, I would tone a bit down the emphasis on the cultural vs natural or natural vs acquired framework… Again, we learn to do many things that are considered as natural.

12) “The presence of a distinct effect of roughness in even the most musically remote PNG community suggests a universal, or at least non-arbitrary, association between roughness and instability.” P. 14. How about a “consistent perceptual association between roughness and instability”? or something like?

Another reason why I don’t like the cultural/learned perspective. Consider the case of a cubist female portrait. We need to learn how to see it, how to associate those strange geometrical figures to a nose here, an eye there, an asymmetric and misplaced breast. After “learning”, the portrait will become clearer. However, we will only have learnt how to decipher it, but the portrait will still seem an uncommon, weird way of representing a female body. Something similar might happen with music, with cultural learning that might facilitate the understanding of serialism or alea, but will not make it sound like perfect consonances...

Reviewer #2: The paper is really well written, the topic is relevant, and the experiments and statistical analyses are sound. My only serious concern is the proxy between dissonance and "sensation of closure".

Major

In my view the question of whether c/d is universal or culture-dependent can be considered as two problems: 1) whether the perception of c/d is universal; and 2) whether the associations of X with c/d are universal. I think most intercultural studies in the past have tackle the second point, and specifically whether pleasantness is universally associated to consonance. In my view, the current paper studies whether the feeling of cloure is universally associated to dissonance (Figure 3 suggests it is not); and whether certain markers that have been related to dissonance in the past (roughness, harmonicity, ...) are universatlly associated to this feeling of closure (Table 1 suggests roughness is). I think these results are relevant and, to my eyes, they look quite robust. On the other hand, I disagree that this can be interpreted as "evidence for a universal association of auditory roughtness with musical dissonance". This conclusion depends on the implicit assumption that dissonance = "feeling of closure"; however, the demonstration that the feeling of closure is not universally associated with dissonance of Fig. 3 seems to contradict this implicit assumption.

Another potential problem is the implicit association of "stability" with "feeling of closure". I think the problem here is that "stability" can be interpreted in two ways: 1) whether the sound is static; 2) whether the sound can be represented stable point (in the sense of dynamic systems). I can see why in the second interpretation a stable sound is intrinsically more likely to feel like a finishing sound than an unstable sound; however, I think this is not necessarily the case in the first interpretation (even if this happen to be the case in Western music). I have the impression that the association of "unstability" with "dissonance" stems rather from the beating sensation of the dissonant chords, which would correspond to the first interpretation.

In my view, the report would be more rigorous avoiding these implicit associations: namely, just stating that "closure/finishing sensation" rather than "dissonance" or "stability" is associated to roughness; and removing the implicit associations between one and the other in the MS. Critically, this would affect not only the abstract, introduction, and discussion, but most importantly also the title.

Minor

I have the impression the paper is very verbose explaining the Valley, the travels, and the presentation procedure, but pushed all the technical methods to the appendix. I would suggest to ensure that the methods section contains at least all the information necessary to reproduce the analyses. I think pushing part of the procedures and the cultural description to the appendix (more generally, shortening the paper) will increase the chances that hardcore psychophysics researchers (which I reckon will be the most interested in the results) read the paper, but of course that is up to the authors to decide.

I am not familiar with Bayesian multilevel logistic regression, so the following questions should be understood as "more details are needed for unfamiliar readers to fully understand the methodology" is they are misguided:

1) Is the evidence ratio really the quotient between the posteriors? To my understanding the evidence is p(data|hypothesis), whereas the posterior is p(hypothesis|data). If the priors for all hypotheses are the same, the ratio between two evidences equals the ratio between two posteriors, but the definition is the former, no? Is not the ratio of the evidences simply Bayes' K factor? As far as I understand K factors are the standard way to report Bayesian results, wouldn't it be better to adhere to this standard notation?

2) How are the priors defined?

3) Is not K>39 a really conservative threshold? I always understood K>sqrt(10) was roughly comparable to p<0.05. I think quantitatively transferring this to classical stats is complicated, since classical analyses can only measure p(data|null). However, assuming there are only two models in the analyses, wouldn't K>39 be the equivalent of p(null) = 0.025 and p(alternative) = 0.975?

4) Why is necessary to specify a range of values for which the effect is practically equivalent to zero? Shouldn't this depend on the variance of the estimates?

5) Since your two models (effect != 0, and effect=0) are mutually exclussive, I would expect the posteriors to sum up to one; why is this not the case? I can imagine why this may be a consequence of the choice of the interval [-.036,+.036]; however: is there not a better way to estimate the likelihood that the data is sampled from an effect distribution with zero mean, incorporating the empirical estimation of the variance?

Potential typos:

Second sentence in Methods: Participants from the Uruwa River Valley" reads a bit awkward; I would for instance use "participants were excluded".

Page 6, "The cadences and melodies blocks..." this sentence sounds a bit strange. Maybe you could write something like "because we measured ratings of ..."?

Page 7, "When you hear which tune, which one is finished" is this correct?

Page 8, "mdl_cut_int_triad_1" not listed in Table 5, maybe a typo?

Minor suggestions:

Tables could use self-contained explanatory captions.

Tables and figure in the appendix could be numbered as A1, A2, ...

Machine-print variable names and headings in Table 5 are really untransparent: I would rewrite then in human-readable format. Much less urgent, but maybe still worth consider, in the name of the variables in Table 7

6. PLOS authors have the option to publish the peer review history of their article (what does this mean?). If published, this will include your full peer review and any attached files.

Reviewer #1: No

Reviewer #2: **Yes: **Alejandro Tabas

---

## [Author Response · Author response to Decision Letter 0]

20 Aug 2023

Please see the attached Cover Letter & Response to Reviewers pdf

---

## [Decision Letter · Decision Letter 1]

4 Sep 2023

Evidence for a universal association of auditory roughness with musical stability

PONE-D-23-12398R1

Dear Dr. Milne,

We’re pleased to inform you that your manuscript has been judged scientifically suitable for publication and will be formally accepted for publication once it meets all outstanding technical requirements.

Kind regards,

Jessica Adrienne Grahn

Academic Editor

PLOS ONE

Additional Editor Comments (optional):

Reviewers' comments:

Reviewer's Responses to Questions

**Comments to the Author**

1. If the authors have adequately addressed your comments raised in a previous round of review and you feel that this manuscript is now acceptable for publication, you may indicate that here to bypass the “Comments to the Author” section, enter your conflict of interest statement in the “Confidential to Editor” section, and submit your "Accept" recommendation.

Reviewer #1: All comments have been addressed

Reviewer #2: All comments have been addressed

2. Is the manuscript technically sound, and do the data support the conclusions?

Reviewer #1: Yes

Reviewer #2: Yes

3. Has the statistical analysis been performed appropriately and rigorously? 

Reviewer #1: Yes

Reviewer #2: Yes

4. Have the authors made all data underlying the findings in their manuscript fully available?

Reviewer #1: Yes

Reviewer #2: Yes

5. Is the manuscript presented in an intelligible fashion and written in standard English?

Reviewer #1: Yes

Reviewer #2: Yes

6. Review Comments to the Author

Reviewer #1: Dear authors, thank you for the effort made in improving your manuscript, which now looks to my eyes ready to be published.

One very small thing: p. 2, line 61, you label Y as the "affective response", I would probably label it as "subjective response" given that such responses are not necessarily affective (e.g., stability), but rather perceptual (or something like) in nature.

Reviewer #2: The authors have provided for a thorough revision of the MS, addressed my concerns when they were grounded, and convincingly argued against them when they were not.

7. PLOS authors have the option to publish the peer review history of their article (what does this mean?). If published, this will include your full peer review and any attached files.

Reviewer #1: No

Reviewer #2: **Yes: **Alejandro Tabas

---

## [Editor Report · Acceptance letter]

11 Sep 2023

PONE-D-23-12398R1 

Evidence for a universal association of auditory roughness with musical stability 

Dear Dr. Milne:

I'm pleased to inform you that your manuscript has been deemed suitable for publication in PLOS ONE. Congratulations! Your manuscript is now with our production department. 

Kind regards, 

on behalf of

Dr Jessica Adrienne Grahn 

Academic Editor

PLOS ONE